# Acute Lung Functional and Airway Remodeling Effects of an Inhaled Highly Selective Phosphodiesterase 4 Inhibitor in Ventilated Preterm Lambs Exposed to Chorioamnionitis

**DOI:** 10.3390/ph16010029

**Published:** 2022-12-26

**Authors:** Matthias Christian Hütten, Tim Brokken, Helene Widowski, Tobias Monaco, Jan Philipp Schneider, Markus Fehrholz, Daan Ophelders, Boris W. Kramer, Steffen Kunzmann

**Affiliations:** 1Neonatology, Pediatrics Department, Maastricht University Medical Center, 6229 HX Maastricht, The Netherlands; 2GROW—School for Oncology and Reproduction, Faculty of Health, Medicine and Life Sciences, Maastricht University Medical Center, 6200 MD Maastricht, The Netherlands; 3Department of BioMedical Engineering, Maastricht University Medical Center, 6200 MD Maastricht, The Netherlands; 4Emergency Department, University Hospital Düsseldorf, Heinrich Heine University, 40225 Düsseldorf, Germany; 5Hannover Medical School, Institute of Functional and Applied Anatomy, Carl-Neuberg-Straße 1, 30625 Hannover, Germany; 6Biomedical Research in Endstage and Obstructive Lung Disease Hannover (BREATH), Member of the Ger-man Center for Lung Research (DZL), Hannover Medical School, Carl-Neuberg-Straße 1, 30625 Hannover, Germany; 7Monasterium Laboratory, Skin and Hair Research Solutions GmbH, 48149 Muenster, Germany; 8University Children’s Hospital Würzburg, University of Würzburg, 97080 Würzburg, Germany; 9Division of Obstetrics and Gynecology, School of Women’s and Infants’ Health, University of Western Australia, Crawley, WA 6009, Australia; 10Neuroplast BV, 6167 RD Geleen, The Netherlands; 11Clinic of Neonatology, Bürgerhospital Frankfurt am Main, 60318 Frankfurt, Germany

**Keywords:** PDE inhibitors, neonatal lung injury, respiratory distress syndrome, bronchopulmonary dysplasia

## Abstract

Phosphodiesterase (PDE) inhibition has been identified in animal studies as a new treatment option for neonatal lung injury, and as potentially beneficial for early lung development and function. However, our group could show that the inhaled PDE4 inhibitor GSK256066 could have dose-dependent detrimental effects and promote lung inflammation in the premature lung. In this study, the effects of a high and a low dose of GSK256066 on lung function, structure and alveolar development were investigated. In a triple hit lamb model of Ureaplasma-induced chorioamnionitis, prematurity, and mechanical ventilation, 21 animals were treated as unventilated (NOVENT) or 24 h ventilated controls (Control), or with combined 24 h ventilation and low dose (iPDE1) or high dose (iPDE10) treatment with inhaled GSK 256066. We found that high doses of an inhaled PDE4 inhibitor impaired oxygenation during mechanical ventilation. In this group, the budding of secondary septae appeared to be decreased in the preterm lung, suggesting altered alveologenesis. Ventilation-induced structural and functional changes were only modestly ameliorated by a low dose of PDE4 inhibitor. In conclusion, our findings indicate the narrow therapeutic window of PDE4 inhibitors in the developing lung.

## 1. Introduction

The developing lung of preterm infants is prone to perinatal lung injury [1]. Prenatal insults such as chorioamnionitis and postnatal insults, e.g., mechanical ventilation for treatment of neonatal respiratory distress syndrome (RDS), are known risk factors for neonatal lung injury and can promote the development of bronchopulmonary dysplasia (BPD) [2].

Therapeutic strategies for neonatal lung injury are however scarce, mainly due to the multi-factorial origin of the disease [3]. Phosphodiesterase (PDE) inhibition has been identified as a new treatment option in animal studies of RDS, neonatal lung injury, and BPD [4]. PDEs are a group of enzymes that catalyze the hydrolysis of phosphodiester bonds of 3′,5′cylic adenosine (cAMP) or guanosine monophosphate (cGMP) [4]. Unselective PDE inhibitors such as theophylline induce bronchodilatation and provide amongst others anti-inflammatory effects [4]. Caffeine, which has weak unselective inhibitory effects on PDE, is widely used for the treatment of apnea of prematurity and has been clinically shown to decrease the incidence of BPD [5]. PDE4 is an isoform that has major activity in lung and inflammatory cells, where it reduces the cleavage of cAMP [6]. Selective inhibition of the PDE isoenzyme 4 has so far mainly been tested in pulmonary diseases of adulthood, such as asthma [7], chronic obstructive lung disease (COPD) [8], and other non-COPD respiratory diseases [9].

Three main modes of action have been described which make pulmonary PDE inhibition promising in the context of neonatal RDS and BPD. First, PDE inhibition has been shown to decrease pulmonary inflammation in various animal models of hyperoxia-induced BPD [10], meconium aspiration syndrome [11,12] and lavage-induced lung injury [13]. In a recent study on mice exposed to lipopolysaccharide, PDE4 inhibitor Ibudilast ameliorated acute RDS by alleviating inflammation [14]. In a rodent lung lavage model, an intratracheally administered PDE4 inhibitor prevented further hyaline membrane formation and infiltration of neutrophil leukocytes [15].

Second, in these animal models, improvement of lung function by different PDE inhibitors has been described, mainly improvement of oxygenation [4,11,13,16,17]. In the context of mechanical ventilation and acute lung injury (ALI) initiated by lung lavage, systemic iPDE4 treatment had beneficial effects on both oxygenation and ventilation parameters, such as oxygenation index (OI), Horowitz score (PF), alveolar-arterial gradient (AAG) and ventilation efficacy index (VEI), on lung edema [13], and on lung compliance when combined with surfactant lavage in a meconium-aspiration syndrome (MAS) model [18]. When combined with a recombinant surfactant protein C-based surfactant, intratracheal PDE4 inhibitor augmented positive surfactant effects on oxygenation [15].

Third, tissue remodeling processes in the lung are modulated by selective PDE4 inhibition. In vitro, PDE4 inhibition modulated transforming growth factor (TGF)-β signaling, which has been identified as an important factor in perinatal lung remodeling processes [19]. However, rodent models of hyperoxia-induced BPD showed conflicting results of PDE inhibition with regard to alveolarization. While some authors described partial or complete improvement of alveolarization disturbed by hyperoxia [10,20,21], others found no preventive effect of PDE4 inhibitors on hyperoxia-induced impairment of alveolarization, and even deleterious effect in animals exposed to normoxia [22].

In our study, we tested an inhalative PDE4 inhibitor (GSK256066) in a triple-hit lamb model of Ureaplasma-induced chorioamnionitis, prematurity and mechanical ventilation. The preterm sheep model is well established to investigate injury of the preterm lung related to ventilation [23], due to its physiological and anatomical similarities compared to the human situation [24,25], representing a clinical setting of neonatal intensive care patients. Regarding inflammation, our group previously described in our model that a low dose of the inhalative PDE4 inhibitor delayed but not prevented ventilation-induced lung inflammatory response, while a high dose even increased pulmonary inflammation and was associated with significant lung injury [26].

Because of these observations and the known small therapeutic range of PDE4 inhibitors [27], we hypothesized that treatment with a low dose but not a high dose of an inhalative PDE4 inhibitor might result in beneficial effects on lung function and pulmonary structure in our model.

## 2. Results

### 2.1. Effect of PDE4 Inhibition on Lung Function Parameters

Ventilatory parameters changed significantly over time in all ventilated groups (Figure 1a–f). The oxygenation index (OI) was significantly lower in the iPDE1 group compared to the iPDE10 group. High-dose treated animals showed a trend towards increased OI compared to control animals (*p* = 0.056, Figure 1a), and a trend towards increased alveolar-arterial gradient (AAG) (*p* = 0.063 between groups, *p* = 0.101 vs. control, Figure 1b). Horowitz score (PF), alveolar partial pressure of oxygen (P_A_O_2_), ventilator efficiency index (VEI) and compliance were not significantly altered (Figure 1c–f).

### 2.2. Effect of PDE4 Inhibition on Static Lung Compliance and Lung Water Content

Static compliance measured as post mortem pressure-volume curve normalized for body weight showed increased lung volumes in ventilated animals; however, post hoc analysis revealed that this increase was only significant in animals ventilated without PDE inhibitor treatment (* *p* < 0.05 vs. NOVENT, Figure 2). Wet/dry ratio as a marker of pulmonary edema was significantly decreased in ventilated control animals compared to non-ventilated animals (* *p* < 0.05, Figure 3), but not in both iPDE treatment groups. The difference in lung weight/bodyweight ratio was not significant between groups (3.0% ± 0.6%, 2.2% ± 0.0%, 2.1% ± 0.2%, 2.0% ± 0.3%, respectively).

### 2.3. Effect of PDE4 Inhibition on Lung Structure

Mean linear intercept (MLI) as a measure of morphometric lung changes determined by the method published by Crowley et al. [28] was significantly lower in the iPDE1 group compared to ventilated control group animals. We observed a wide spread in the iPDE10 group, indicating an inhomogeneous pattern of mean free distance between pulmonary airway surfaces, possibly due to lung injury (Figure 4).

Lungs from 15 of 21 animals were processed for and analyzed by design-based stereology [29,30]. Lung volumes determined by both Scherle’s method based on Archimedes principle (V_Arch_) [30,31] and the Cavalieri method (V_Cav_) [30,32] normalized for body weight did not differ between groups (V_Arch_ 16.7 ± 0.9 cm^3^/kg, 13.3 ± 1.6 cm^3^/kg, 15.3 ± 1.6 cm^3^/kg, and 15.1 ± 2.1 cm^3^/kg, respectively, V_Cav_ 22.3 ± 2.3 cm^3^/kg, 21.6 ± 2.2 cm^3^/kg, 19.5 ± 2.5 cm^3^/kg, and 21.5 ± 4.6 cm^3^/kg, respectively, given as mean ± SEM), nor did the air-sided surface of the (prospective) alveolar region (S(alv,lung), Table 1).

However, the mean absolute air-sided surface of the (prospective) alveolar region was the largest in the NOVENT group, suggesting a loss of surface in ventilated animals. The lack of significance is probably caused by the small size of the study groups. In all groups of ventilated animals, we found—compared to unventilated animals—statistically significant differences in the mean thickness of tissue in the (prospective) alveolar region (t(tissue)) and the surface density of air-sided surface with differentiated alveolar epithelium in the (prospective) alveolar region related to parenchyma (Sv(alv/par)) as well as to the entire lung (Sv(alv/lung)), but without relevant effects of iPDE treatment (Table 1). The mean linear intercept length of airspace (Lm(air)) calculated from stereological parameters was higher when comparing pooled data from ventilated animals compared to non-ventilated animals, but however not significantly in the group comparison (Table 1). In addition, one animal in the Control group and two animals in the iPDE10 group showed cavities with a short diameter > 3 mm on tissue slabs not being physiological lumina, with a volume fraction related to the lung (Vv(bull/lung)) of 0.04, 0.004 and 0.04, respectively.

### 2.4. Effect of PDE4 Inhibition on Elastin Deposition

The total number of profiles of elastin foci per high power field was significantly lower in all ventilated groups compared to not ventilated controls, without a treatment effect of iPDE (Figure 5a). Elastin foci were localized at both the walls of distal airspaces and the tips of secondary septae. The latter were described as elastin buds, indicating the contours of the future alveoli and therefore alveologenesis [33]. The relative amount of elastin buds localized at the tip of secondary septae was significantly lower in iPDE10 animals compared to unventilated and ventilated controls (Figure 5b,c–f).

### 2.5. Effect of PDE4 Inhibition on TGF-β Signalling

TGF-β mRNA levels were significantly increased in ventilated controls, but not in iPDE treated animals compared to unventilated animals (Figure 6a). Connective tissue growth factor (CTGF) mRNA levels were increased in all ventilated groups compared to unventilated controls, however significantly only in the control group and iPDE10 group animals (Figure 6b).

## 3. Discussion

In our model, ventilation had a clear effect on lung function parameters. However, treatment of ventilated animals with a low dose of an inhaled PDE4 inhibitor did not significantly ameliorate these changes compared to controls. In contrast, inhalation with a high dose of the PDE4 inhibitor impaired oxygenation. Although we did not find differences in dynamic compliance, static compliance depicted by post-mortem pressure-volume measurement was only significantly increased in ventilated animals not receiving iPDE treatment, indicating slightly worse static compliance after iPDE treatment. In line with previously published findings from our group on histological signs of lung injury and lung inflammation in high-dose treated animals [26], wet/dry ratio as a marker of lung edema did only significantly decrease in ventilated control animals compared to the unventilated group, but not in iPDE4 treated animals [26]. Earlier described positive effects of PDE inhibition on lung edema in rodent models were associated with decreased inflammation and increased oxygenation [10,11], which indicates that functional impairment in our high-dose animals is likely to be associated with dose-dependent increased inflammatory processes.

Furthermore, we observed a clear effect of 24 h ventilation on lung morphology, while the effects of PDE4 inhibition were less distinct. In all ventilated study groups the air-sided surface density of the prospective alveolar region with differentiated alveolar epithelium related to parenchyma as well as to the entire lung was significantly decreased compared to the non-ventilated group. In contrast, the absolute surface area as a reference space independent indicator of lung damage or impaired development did not show any significant differences. Thus, one might deduce that the changes in surface density are a result of changes in the reference space, i.e., distension of the lung caused by ventilation compared to non-ventilated lungs. In contrast, elevated tissue thickness in all ventilated groups may indicate impaired development or secondary changes like inflammatory infiltrations. Interestingly, while design-based stereology in a subset of animals showed no ventilation effect on Lm(air), a semi-automated approach including all animals showed significant differences in mean linear intercept in the group comparison between ventilated control animals and low dose PDE4 inhibitor-treated animals, while high-dose treated animals showed a higher inter-individual variability. When comparing these findings to previous studies from the hyperoxia-rodent model of BPD, we find conflicting results. While Visser et al. described a protective effect of PDE4 inhibitor cilomilast on the septal thickness and alveolar edema [10], hyperoxia-induced increase in mean linear intercept was not improved significantly after systemic PDE4 inhibitor treatment in a study using PDE4 inhibitor rolipram [21]. More importantly, Mehats et al. could show in this model that the decrease of alveolar surface area in rolipram-treated pups could be found in both hyperoxia-exposed pups and pups not exposed to an injurious stimulus [22]. In this context, it remains speculative if the observed decrease in the iPDE1 group depicts a beneficial effect, such as a protection against ventilation-induced lung over-distension, or if the observed decrease is a result of beginning secondary parenchymal changes. Because of the broad spread in MLI observed in the iPDE10 group, results from this group cannot help to substantiate our hypothesis. However, due to the semi-automated nature of the method used by Crowley et al. [28], this spread might indicate a non-physiological inhomogeneity in the lung after high-dose treatment.

However, we found indications of lasting developmental impairment. While we interpret the decrease in the total number of elastin profiles in all three ventilated groups as an effect of mechanical ventilation, the significant relative decrease of profiles of elastin buds at the tip of secondary alveolar septae in the high-dose iPDE group suggests impaired alveologenesis [33]. The resulting pattern of a more diffuse elastin expression has earlier been described as a result of exposure of the immature lung to antenatal infection [34], and results in a simplification of lung structure. This is in contrast to data from the hyperoxia rodent model, which indicated a protective effect of PDE inhibition against disturbed alveolarization [10,21]. We hypothesized that the unfavorable distribution of elastin could be determined by modulation of the TGF-β and its downstream mediator CTGF, which both have been linked to BPD development [35]. In previous experiments, TGF-β was upregulated in fetal lambs exposed to intra-uterine inflammation and short postnatal ventilation [36]. CTGF has been associated with remodeling and fibrosis both in ventilation-mediated injury and inflammation [35]. In our study, both TGF-β and CTGF mRNA were increased in ventilated lambs compared to unventilated animals, and CTGF mRNA in animals treated with the high dose of iPDE. However, in the low-dose group, this increase was ameliorated. For systemic iPDE4 treatment, it has been shown that rolipram could antagonize TGF-β induced Smad signaling and TGF-β regulated CTGF [19]. Our data indicate that this positive effect can be overthrown by a high dose of PDE4 inhibitor.

Our study is limited by the fact that dose-finding is species dependent. PDE4 inhibitors are known to have a narrow therapeutic range. Although we based the high dose on previous work in rodents and the low dose on clinical studies in humans, the observed negative effects can possibly be attributed to an over-inhibition of PDE4. An even lower dose over a longer period could possibly prevent this over-inhibition. Another limitation arises from the limited experimental groups due to animal ethical reasons. We, e.g., did not include a group without chorioamnionitis in our study, and therefore cannot rule out that the interaction of antenatal inflammation with postnatal treatment played a crucial role. However, due to the frequent presence of chorioamnionitis in preterm babies and its known effects on pulmonary outcomes [37], the model we chose is near to the clinical situation. Ethical reasons also limited the number of animals per group, which was for one analysis in this study (design-based stereology) further limited to a subset of animals because of limitations in the available material.

In conclusion, low-dose treatment with an inhalative PDE4 inhibitor did ameliorate some ventilation-induced changes in the preterm lamb lung, but without a clear effect on stereological parameters and elastin deposition. In contrast, high doses of an inhaled PDE4 inhibitor seemed to have not only acute negative effects on oxygenation but may also have lasting negative effects on forming of secondary septa, in line with previously described lung inflammation [26]. Our findings indicate both the narrow therapeutic window of PDE4 inhibitors in the developing lung, and the need for further research in the approach to modulate lung inflammation, support developmental processes in the preterm lung, and prevent BPD.

## 4. Materials and Methods

### 4.1. Animal Study

The animal experimental study protocol was approved by the institutional Animal Ethics Research Committee, Maastricht University, The Netherlands, and has previously been published [26]. In short, 21 date-mated ewes underwent ultrasound-guided intraamniotic injection of Ureaplasma parvum HPA 5 5 × 10^5^ CCU at 122 d gestational age (GA), and intramuscular injection of betamethasone (12 mg, Celestone^®^, Schering-Plough, North Ryde, New South Wales, Australia) at 128 d GA. Before surgical delivery at 129 d GA, lambs were randomly assigned to four different treatment groups: not ventilated controls (NOVENT), 24 h ventilation without iPDE4 treatment (Control), 24 h ventilation with 12 hourly administration of 1 µg/kg GSK256066 (iPDE1) or 24 h ventilation with 12 hourly administration of 10 µg/kg GSK256066 (iPDE10). During cesarean section, lambs were intubated and equipped with umbilical artery and vein catheters, followed by cord clamping. After weighing, animals in the NOVENT group were immediately sacrificed by a lethal dose of pentobarbital, while animals assigned to one of the three 24 h ventilation groups were transferred to an infant radiator bed (IW930 Series CosyCot™ Infant Warmer, Fisher & Paykel Healthcare, Auckland, New Zealand). Before connecting the ventilator, 200 mg/kg surfactant (Poractant alpha, Curosurf^®^, Chiesi Pharmaceuticals, Parma, Italy) was endotracheally applied, followed by conventional ventilation (synchronized intermittent mandatory ventilation, SIMV), for 24 h under continuous sedation using an infant ventilator (Fabian HFO^®^, Acutronic, Hirzel, Switzerland). All animals received an intravenous single-loading dose of caffeine (20 mg/kg, Peyona^®^, Chiesi Pharmaceuticals, Parma, Italy).

GSK 256066 (Selleckchem, Munich, Germany) was dissolved in 1% DMSO and nebulized via a vibrating membrane nebulizer (eFlow^®^ Neonatal Nebulizer System, PARI Pharma, Munich, Germany) connected between tube and ventilation tubes at 30 min and 12 h of the experimental period. Ventilatory parameters including positive inspiratory pressure (PIP), mean airway pressure (MAP), positive end-expiratory pressure (PEEP), respiratory rate (RR), fraction of inspired oxygen (FiO_2_) and Compliance (C) were regularly recorded, and arterial blood gas analysis was performed using an iStat device (Abbott Point of Care Inc., Abbott Park, IL, USA).

Based on these data, the following parameters were calculated:Ventilator efficiency index (VEI) = 3800/(RR × 0.736 × (PIP − PEEP) × pCO_2_),
Horowitz score (PF) = pO_2_/FiO_2_
Oxygenation index (OI) = FiO_2_ × 0.736 × MAP/pO_2_
Alveolar partial pressure of oxygen (P_A_O_2_) = (FiO_2_ × (P_atm_ − P_H2O_)) − (pCO_2_/RQ), with the assumption that P_atm_ = 760 mmHg, P_H2O_ = 47 mmHg, and respiratory quotient (RQ) = 0.8
Alveolar-arterial gradient (AAG) = P_A_O_2_ (mmHg) − pO_2_ (mmHg)

At the end of the experiment, animals were euthanized by an overdose of pentobarbital. The thorax was opened, and the tube was disconnected from the ventilator and inflated to a maximum pressure of 40 cm H_2_O. Deflation gas volumes were recorded and normalized for body weight as described before [38]. The lungs were removed from the chest and weighed separately. The right upper lobe (RUL) was inflation-fixed in 10% buffered formaldehyde for 24 h. Paraffin-embedded RUL sections (4 µm) were used for microscopic analyses. In 15 of 21 animals, the left lung was additionally prepared for stereology by inflation-fixation with 1.5% formaldehyde/1.5% glutardialdehyde in 0.15 M HEPES [29,39]. Tissue samples from the right middle lobe (RML) were snap-frozen. A piece of the right lung was put in an open aluminum pot, weighed, and subsequently incubated in dry air at 37 °C. Weight was obtained daily until stable values were reached, and the wet/dry ratio was calculated.

### 4.2. Microscopic Analysis

Deparaffinized sections of the RUL (4 µm) were stained with hematoxylin and eosin. MLI was determined by semi-automated measurement as published by Crowley et al. [28]. H&E slides were scanned and imported in Qpath (Bioimage analysis software; version 0.3.2). Ten randomly placed annotations were made at a magnification of 200×. Blood vessels and bronchioles were excluded from all annotations. Subsequently, the annotations were exported to Fiji ImageJ (version 2.3.051), an open-source image processor. A plugin created by Crowley et al. was installed, and an area of 200 square pixels per point was used to create gridlines [28]. Finally, the plugin was run for the whole project, and the MLI was determined by calculating the mean length per animal. Lines < 5 µm were excluded because they represent mostly inter-alveolar tissue.

### 4.3. Stereological Analysis

Total lung volume was determined by Scherle’s method [31] in water. Three measurements were taken to calculate the arithmetic mean [39]. Samples were generated according to systematic uniform random sampling (SURS) [29,30,40] and for later lung volume estimation based on the Cavalieri principle [25,27], the samples were photographed with a scale. After temporary storage in the primary fixative, the samples were postfixed in 1%-osmium tetroxide in cacodylate buffer and 1% uranyl acetate in water, dehydrated in an ascending series of acetone and embedded in glycol methacrylate (Technovit^®^ 8100, Heraeus, Wehrheim, Germany) (cf. [39]). 1.5 µm thick sections, stained with toluidine blue were digitized by a slide scanner (Axio Scan Z.1, Zeiss, Jena, Germany, pixel size 0.22 µm/pixel) for stereological analysis.

Stereological analysis was conducted in a cascade design using point and intersection counting and associated calculations [29]. The analysis was performed on both a macroscopic level as well as different (virtual) microscopic levels. For macroscopic evaluation, the sample images were cropped using Fiji [41] and analyzed using the STEPanizer [42]. The microscopic analysis was performed in the newCAST^TM^ module of the software package Visiopharm Integrator System^TM^, v5.3.1640 (Visiopharm, Hoersholm, Denmark). For microscopic evaluation, the total section area per lung was estimated in the first step by point counting to estimate the needed sampling fraction at higher magnifications in order to generate enough counting events. Parenchyma was defined as the alveolar (or prospective alveolar) region, i.e., airspaces lined by alveolar epithelial type 1 and 2 cells, and the tissue in between, i.e., mature interalveolar septa or developing mesenchyme in the canalicular or saccular phase of development. The peripheral margins of fine non-parenchyma structures lined by alveolar epithelium (cf. [43]) as well as cellular infiltrations or atelectasis of the (prospective) alveolar region were included in parenchyma. Parameters were reported as follows: V (air,lung) absolute volume of air spaces of the (prospective) alveolar region in the lung, V (tissue,lung) absolute volume of tissue of the (prospective) alveolar region in the lung, S (air,lung) absolute surface of air-sided surface of the (prospective) alveolar region with differentiated alveolar epithelium, Vv(non_cNP/lung) volume fraction of non-coarse non-parenchyma related to the lung, Vv (cNP/lung) volume fraction of coarse non-parenchyma related to the lung, Vv(fNP/non_cNP) volume fraction of fine non-parenchyma related to non-coarse non-parenchyma, Vv(air/par) volume density of air spaces of the (prospective) alveolar region related to parenchyma, Vv (tissue/par) volume density of tissue of the (prospective) alveolar region related to parenchyma, Sv(alv/par) surface density of the air-sided surface of the (prospective) alveolar region with differentiated alveolar epithelium related to parenchyma, Vv(air/lung) volume density of air spaces of the (prospective) alveolar region related to the lung, Vv(tissue/lung) volume density of tissue of the (prospective) alveolar region related to the lung, Sv(alv/lung) surface density of the air-sided surface of the (prospective) alveolar region with differentiated alveolar epithelium related to the lung, Lm(air) mean linear intercept length of air spaces of the (prospective) alveolar region, t(tis) mean thickness of tissue of the (prospective) alveolar region.

### 4.4. Elastin

Sections of the RUL (4 µm) were deparaffinized in an ethanol series and incubated for 20 min in Weigerts Resorcine-Fuchsine (Chroma, 2E 030) at 60–70 °C. After rinsing with water, sections were incubated in tartrazine solution for three minutes at room temperature, washed and dehydrated in ethanol and xylol. Elastin foci could be identified at the distal respiratory unit walls and the tips of secondary septae, and their profiles were counted separately in five pictures per animal at a magnification of 200× using ImageJ software (Rasband, W.s., ImageJ, U.S. National Institute of Health, Bethesda, MD, USA).

### 4.5. RNA Extraction and Real-Time PCR

Total RNA was isolated from lung samples using NucleoSpin^®^ RNA Kit (Macherey-Nagel, Dueren, Germany), in accordance with the manufacturer’s protocol. A Qubit^®^ 2.0 Fluorometer (Thermo Fisher Scientific, Waltham, MA, USA) was used for quantification of total RNA, which was eluted in 60 µL nuclease-free H_2_O (Sigma-Aldrich, St. Louis, MO, USA) and stored at −80 °C until reverse transcription. For RT-PCR, 1 µg of total RNA was reverse transcribed using High Capacity cDNA Reverse Transcription Kit (Thermo Fisher Scientific), in accordance with the manufacturer’s instructions. First-strand cDNA was diluted 1:10 with deionized, nuclease-free H2O (Sigma-Aldrich) and stored at −20 °C until analysis. For quantitative detection of mRNA, 10 µL of diluted first strand cDNA were analyzed in duplicates of 25 µL reactions using 12.5 µL iTaq™ Universal SYBR^®^ Green Supermix (Bio-Rad Laboratories, Hercules, CA, USA), 0.5 µL deionized H_2_O, and 1 µL of a 10 µM solution of forward and reverse primers (Sigma-Aldrich). Levels of mRNA were measured for TGF-β and its downstream mediator CTGF. Primers were ovTGFBfwd 5′-GGCGACCCACAGAGAGGAAATAG-3′, ovTGFBrev 5′-AGGCAGAAATTGGCGTGGTAGC-3′, ovCTGFfwd 5′-ACGGCGAGGTCATGAAGAAGAACA-3′, ovCTGFrev 5′-TGGGGCTACAGGCAGGTCAGTG-3′. PCRs were performed on an Applied Biosystems^®^ 7500 Real-Time PCR System (Thermo Fisher Scientific) using a 2-step PCR protocol after an initial denaturation at 95 °C for 10 min with 40 cycles of 95 °C for 15 s and 60 °C for 1 min. At the end of each run, a melt curve analysis was performed to verify single PCR products. Levels of mRNAs were normalized to those of β Actin. Mean fold changes in mRNA expression were calculated by the ΔΔCT method by Livak and Schmittgen [44].

### 4.6. Statistical Analysis

Statistical data analysis was performed using IBM^®^ SPSS Statistics for Windows, Version 20.0 (IBM Corp., Armonk, NY, USA). Data were depicted as the mean and standard error of means (SEM), and graphs were drawn with GraphPad Prism^®^ v5.0 (GraphPad Software Inc., San Diego, CA, USA). Results from necropsy, microscopy, stereology and RT-PCR were analyzed using One-way analysis of variance (ANOVA) with Bonferroni post hoc testing. For analysis of functional data, repeated measurements ANOVA was performed. Significance was accepted at *p* < 0.05.

## Figures and Tables

**Figure 1 pharmaceuticals-16-00029-f001:**
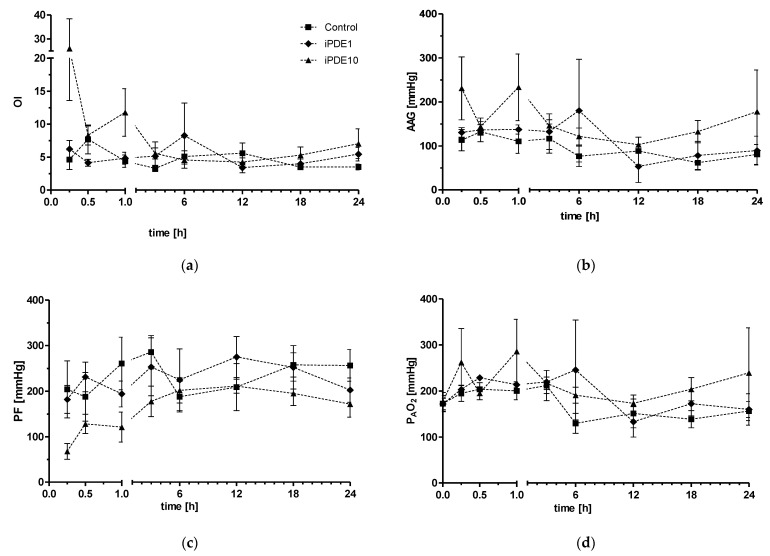
Parameters of oxygenation, including (**a**) Oxygenation index (OI), (**b**) alveolar-arterial gradient (AAG), (**c**) Horowitz score (PF), (**d**) alveolar partial pressure of oxygen (P_A_O_2_), and parameters of ventilation including (**e**) compliance, (**f**) ventilator efficiency index (VEI), during the course of the experiment in animals in control group (■), iPDE1 group (♦) and iPDE10 group (▲), given as mean ± SEM.

**Figure 2 pharmaceuticals-16-00029-f002:**
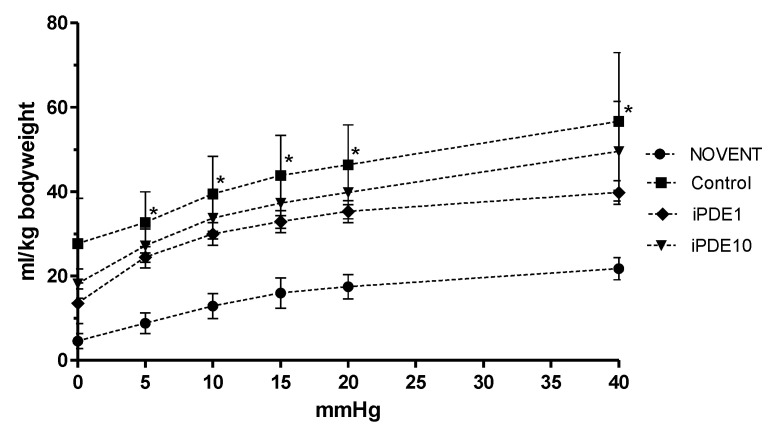
Normalized pressure-volume curve in animals of the NOVENT group (●), control group (■), iPDE1 group (♦) and iPDE10 group (▼), given as mean ± SEM (* *p* < 0.05 vs. NOVENT).

**Figure 3 pharmaceuticals-16-00029-f003:**
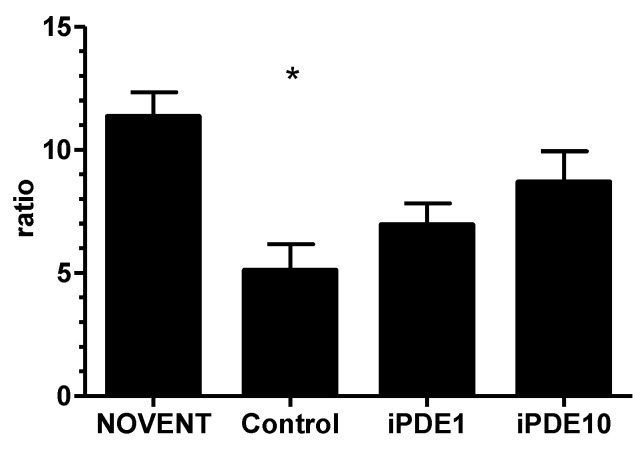
Wet/dry-ratio per group (mean ± SEM, * *p* < 0.05 vs. NOVENT).

**Figure 4 pharmaceuticals-16-00029-f004:**
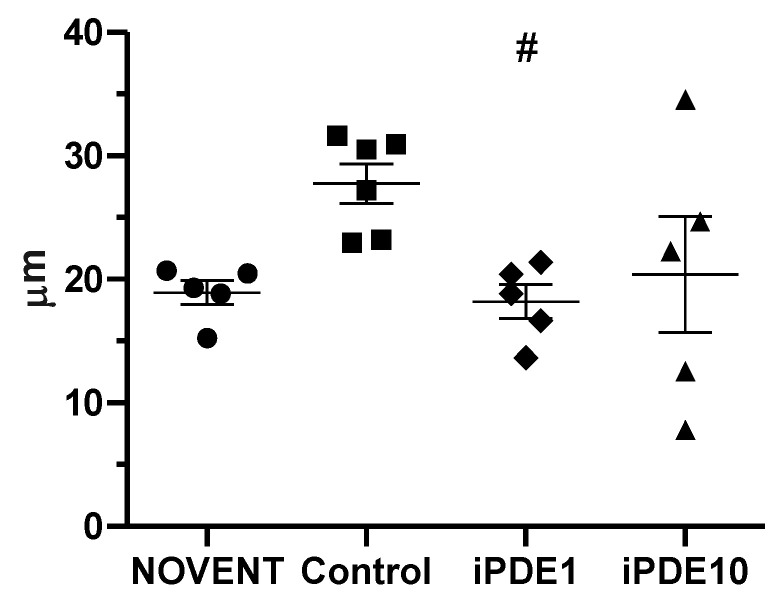
MLI in µm was significantly lower in iPDE1 animals compared to control (^#^
*p* < 0.05 vs. control), whereas the iPDE10 group showed a high individual spread within this group.

**Figure 5 pharmaceuticals-16-00029-f005:**
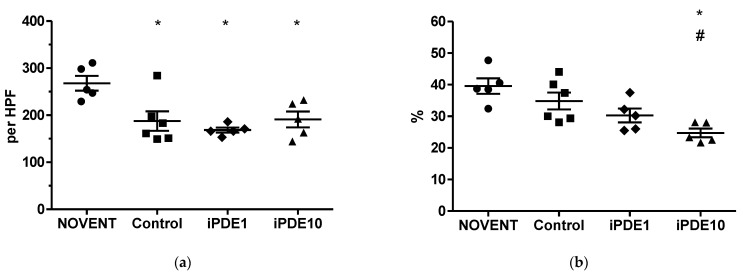
(**a**) Total number of profiles of elastin foci per HPF and (**b**) relative amount of profiles of elastin foci located at the budding secondary septae (* *p* < 0.05 vs. NOVENT, ^#^
*p* < 0.05 vs. control). Example images at 400× magnification for (**c**) NOVENT, (**d**) Control, (**e**) iPDE1 and (**f**) iPDE10.

**Figure 6 pharmaceuticals-16-00029-f006:**
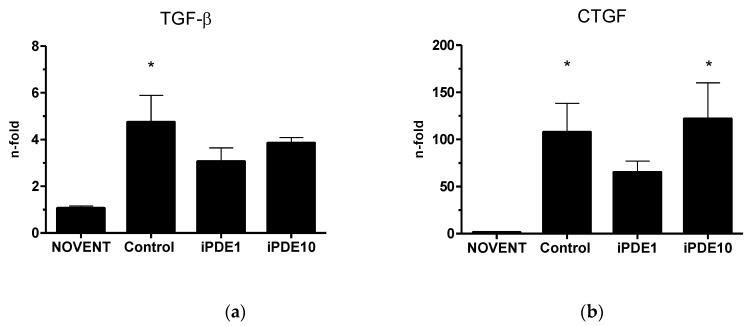
(**a**) TGF-β and (**b**) CTGF mRNA levels relative to unventilated controls (mean ± SEM, * *p* < 0.05 vs. NOVENT).

**Table 1 pharmaceuticals-16-00029-t001:** Stereology of the lung in 15 of 21 animals (data given as mean ± SEM, * *p* < 0.05 vs. NOVENT).

	NOVENT	Control	iPDE1	iPDE10
V_Arch_ (air, lung) [cm^3^]	46.1 ± 8.8	47.9 ± 6.6	37.2 ± 9.4	39.1 ± 13.6
V_Arch_(tissue, lung) [cm^3^]	6.52 ± 2.06	11.79 ± 1.63	8.53 ± 2.49	9.23 ± 2.11
S_Arch_(alv, lung) [m^2^]	4.8 ± 1.0	3.6 ± 0.9	3.0 ± 0.8	2.9 ± 0.7
V_Cav_ (air, lung) [cm^3^]	44.4 ± 7.9	45.2 ± 5.9	34.6 ± 8.3	38.0 ± 12.4
V_Cav_(tissue, lung) [cm^3^]	6.26 ± 1.93	11.09 ± 1.40	7.89 ± 2.13	9.05 ± 1.91
S_Cav_(alv, lung) [m^2^]	4.7 ± 1.0	3.4 ± 0.8	2.8 ± 0.6	2.9 ± 0.6
Vv(non-cNP/lung)	0.98 ± 0.00	0.98 ± 0.01	0.99 ± 0.01	0.97 ± 0.01
Vv(cNP/lung)	0.02 ± 0.00	0.02 ± 0.01	0.01 ± 0.01	0.03 ± 0.01
Vv(fNP/non-cNP)	0.12 ± 0.02	0.13 ± 0.02	0.25 ± 0.06	0.24 ± 0.03
Vv(air/par)	0.88 ± 0.02	0.80 ± 0.01	0.81 ± 0.04	0.80 ± 0.03
Vv(tissue/par)	0.12 ± 0.02	0.20 ± 0.01	0.19 ± 0.04	0.20 * ± 0.03
Sv(alv/par) [cm^−1^]	909.7 ± 29.1	615.5 * ± 36.1	661.1 * ± 69.4	639.6 * ± 57.1
Vv(air/lung)	0.77 ± 0.03	0.68 ± 0.02	0.60 ± 0.05	0.59 ± 0.05
Vv(tissue/lung)	0.10 ± 0.02	0.17 ± 0.01	0.14 ± 0.04	0.15 ± 0.02
Sv(alv/lung) [cm^−1^]	789.1 ± 26.4	521.5 * ± 28.1	492.5 * ± 75.4	468.7 * ± 21.5
Lm(air) [µm]	39.0 ± 2.2	53.1 ± 3.6	50.6 ± 6.9	50.8 ± 6.2
t(tissue) [µm]	2.5 ± 0.4	6.6 * ± 0.5	5.6 * ± 0.5	6.4 * ± 0.4

## Data Availability

Experimental data is contained within the article.

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
