# Peer review of "Acute Lung Functional and Airway Remodeling Effects of an Inhaled Highly Selective Phosphodiesterase 4 Inhibitor in Ventilated Preterm Lambs Exposed to Chorioamnionitis"

_pharmaceuticals, 2022, doi:10.3390/ph16010029_

Round 1

Reviewer 1 Report

The study "Acute lung functional and airway remodeling effects of an in- 2 haled highly selective phosphodiesterase 4 inhibitor in venti- 3 lated preterm lambs exposed to chorioamnionitis" by Matthias et al is well planned and executed. The manuscript is well written and discussed. The study will be  valuable for scientific community. The manuscript can be considered for publication in its current form.

Author Response

We thank the reviewer for the thorough review of our manuscript and the positive and encouraging feedback.

Reviewer 2 Report

I have the following concerns

-Revise the English,

-Insert the symbolic representation in order to differentiate the graphs

-Do the authors have ethical  permission

-Expend the Introduction 

Author Response

We thank the reviewer for the thorough review of our manuscript and the valuable comments. As suggested, the introduction has further been expended in order to provide more information on PDE inhibition and treatment options in neonates. We elaborate on the mechanisms of action of PDE inhibition, and give a more detailed insight into treatment effects in previously published research in other models of neonatal lung injury. We further elaborate on the advantages of our model, and additional references were included. Symbol representation has been added both in the figures and in the legend text to enhance differentiation of graphs. The ethical approval has been obtained and is mentioned both in the methods section and in the Institutional Review Board Statement. English language has been revised, and a spelling check has been performed in MS Word. 

Reviewer 3 Report

Summary:

Phosphodiesterase (PDE) inhibition is a new treatment alternative for neonatal lung injury, potentially beneficial for early lung development and function in animal studies. The authors hypothesized that treatment with inhalative PDE4 inhibitor would improve lung function and pulmonary structure in a triple-hit lamb mode. They found that high doses of an inhaled PDE4 inhibitor impaired oxygenation during mechanical ventilation. The authors conclude that narrow therapeutic window of PDE4 inhibitors in the developing lung.

General concerns:

1.     The treatment effect of PDE inhibition revealed variable results. iPDE exhibited better effects on pressure-volume cure and worse effects on wet/dry-ratio although the differences were not statistically significant. Please clarify these controversial results.

2.     Mean linear intercept (MLI) is interpreted as the mean free distance between gas exchange surfaces in the acinar airway complex. MLI is with emphysematous changes to lung injury. The decreased elastin buds at the tip of secondary alveolar septae indicate impaired alveologenesis. In this study, the control group exhibited higher MLI and lower elastin total number of profiles of elastin foci compared to NOVENT group in Figure 4 and Figure 5, respectively. These results indicate that the control group did not improve lung development and do not support the control group has beneficial effects in Figures 2 and 3. Please clarify these controversial results.

3.     The manuscript lacks a statistical analysis section.

Author Response

We thank the reviewer for the thorough reveiw of our manuscript and the valuable comments. As suggested, we would like to clarify the interpretation of our results. Regarding the changes found in the pressure-volume curve, we hypothesize that the increase was an effect of mechanical ventilation, which animals received in the Control, PDE1 and PDE 10 group, whereas in contrast the NOVENT group did not receive ventilation but was sacrificed directly after birth, before breathing. Interestingly, the increase was only significant in the comparison between Control and NOVENT, suggesting a slightly worse static compliance in both PDE treated groups compared to the only ventilated animals of the control group, although we did not find differences in dynamic compliance. We added additional explanation in the discussion.

We also interpret the findings in MLI as ventilation related. Animals in the control group had a higher mean linear intercepts as NOVENT animals, which in our opinion is related to mechanical ventilation. The significant difference between control and iPDE1 group animals is indeed difficult to interpret: The lower MLI in iPDE1 might on the one hand be a result of a protective effect of low dose inhibition on ventilator induced overextension, on the other hand might be related to secondary parenchymal changes. We discuss this dilemma in our discussion, in the context of previous animal data. Because of the broad spread in MLI observed in the iPDE10, results from this group can however not help to substantiate our hypothesis. However, due to the semi-automated nature of the method used from Crowley et al., this spread might indicate a non-physiological inhomogeneity in the lung after high-dose treatment.  We added an additional part in the discussion. Finally, the decrease in the total number of elastin foci is in our opinion an effect of mechanical ventilation in the control, iPDE1 and iPDE10 group, whereas the profile of the elastin foci is mainly impaired in the iPDE10 group. We extended our discussion accordingly.

Finally, as suggested by reviewer three, we added a statistical analysis section in the methods section.